# Bile Acids: Physiological Activity and Perspectives of Using in Clinical and Laboratory Diagnostics

**DOI:** 10.3390/molecules27227830

**Published:** 2022-11-13

**Authors:** Yaroslav Shansky, Julia Bespyatykh

**Affiliations:** 1Department of Molecular Medicine, Center of Molecular Medicine and Diagnostics, Federal Research and Clinical Center of Physical-Chemical Medicine of Federal Medical Biological Agency, Malaya Pirogovskaya Str., 1a, 119435 Moscow, Russia; 2Department of Expertise in Doping and Drug Control, Mendeleev University of Chemical Technology of Russia, Miusskaya Square, 9, 125047 Moscow, Russia; 3Department of Public Health and Health Care, Federal Scientific State Budgetary Institution «N.A. Semashko National Research Institute of Public Health», Vorontsovo Pole Str., 12-1, 105064 Moscow, Russia

**Keywords:** bile acids metabolism, high-performance liquid chromatography, mass spectrometry, metabolome

## Abstract

Bile acids play a significant role in the digestion of nutrients. In addition, bile acids perform a signaling function through their blood-circulating fraction. They regulate the activity of nuclear and membrane receptors, located in many tissues. The gut microbiota is an important factor influencing the effects of bile acids via enzymatic modification. Depending on the rate of healthy and pathogenic microbiota, a number of bile acids may support lipid and glucose homeostasis as well as shift to more toxic compounds participating in many pathological conditions. Thus, bile acids can be possible biomarkers of human pathology. However, the chemical structure of bile acids is similar and their analysis requires sensitive and specific methods of analysis. In this review, we provide information on the chemical structure and the biosynthesis of bile acids, their regulation, and their physiological role. In addition, the review describes the involvement of bile acids in various diseases of the digestive system, the approaches and challenges in the analysis of bile acids, and the prospects of their use in omics technologies.

## 1. Introduction

There has been an increased interest in omics technologies in the fields of biology and medicine in recent years. The term “omics” commonly refers to technologies that aim toward the achievement of genomics, transcriptomics, proteomics, and metabolomics. According to M. Bennett, metabolomics is “a systematic study of unique chemical ‘fingerprints’ specific to processes occurring in living cells”, that is, the study of their low-molecular metabolic profiles [1]. The focus of metabolomics is on the set of low-molecular weight metabolites of biological transformations in the human organism. This is generally called the metabolome.

Many metabolites contained in the human body exist in very low concentrations (approximately 10^−9^–10^−11^ M), and the introduction of the aforementioned method of analysis into routine practice is required for their systematic content. Because of this, the development of metabolomics was made possible using highly sensitive and specific methods of analysis, such as high-performance liquid chromatography (HPLC), nuclear magnetic resonance spectrometry (NMR), mass spectrometry, etc. [2].

The metabolome depends on the functional state of a living organism. Therefore, studies of the metabolome have been advanced in various branches of medicine, from neonatal screening to toxicology [2,3]. Metabolite profiles are highly specific and useful for diagnosis as well as clinical and laboratory monitoring of therapy. In the process of this research, new functional markers in the human body were also identified. In other words, the metabolome can characterize the human molecular phenotype [4].

The human digestive system involves the conversion of a large number of metabolites. Bile acids (BAs), which are typically known for emulsifying lipids during digestion, have recently captured the interest of a number of researchers. However, with the development of modern methods of analysis, their other functions, connection with other body systems, and participation in metabolic pathways have also been identified [5]. It has been ascertained that BAs differ significantly in their biological characteristics despite their similar chemical structure. These points and approaches to the analysis of the BAs metabolome and their application in medicine will be considered in this review.

The review has a traditional structure and contains seven main sections, in addition to the Introduction. The first section reviews bile acids’ structure and their biosynthesis, along with their regulation. The second section focuses on the physiological functions of bile acids, with their emphasized signaling function, as well as relationships between bile acids and gut microbiota. The third section reviews the pathological conditions and diseases of the gastrointestinal tract which are associated with bile acids according to published studies. The “Approaches to the analysis of bile acids” section is about modern methods of analyzing bile acids in diagnostics, especially using high throughput chromatography and mass spectrometry as selective and sensitive methods for many bioactive molecules. “The metabolomics of bile acids” (earlier “Bile acids in omics technologies”) section describes previously published studies of the whole bile acids metabolome to assess the condition of the human organism with various diseases and during therapy. The section “The perspectives and limitations in medicine” reviews several limitations and difficulties attached to recent studies of the metabolome as well as an outlook of adaptation of actual data to clinical diagnostics. The traditional Conclusions section summarizes all outlined reviewed data and provides a conclusion about the growing role of the bile acids metabolome in clinical and diagnostical medicine.

## 2. The Structure and Metabolism of Bile Acids

### 2.1. The Structure and Physicochemical Properties

Bile acids (BAs) are a group of cholesterol derivatives which are produced during its catabolism. The structure of BAs includes a cholane ring of 24 carbon atoms condensed into 4 rings, a pentanoic chain with COOH-group in the C24 position, and 1–3 hydroxyl groups (Figure 1) [5,6,7,8,9]. The BAs’ structure appears similar to that of steroid hormones and is thus related to their biological activity [10]. BAs differs by the presence of various functional groups in the C3, C6, C7, C12, C18, and C19 positions of the steroid ring. Many of these substituents are residuals of glycine, glucuronic acid, taurine, and sulfonic acid, and are produced in the process of BAs’ biotransformation [8,11,12].

*Cis-trans* isomerism is a characteristic for BAs due to the different orientation of the substituents in the C3-, C7-, and C12-positions, as well as methyl radicals in the C18- and C19-positions. In most cases, hydroxyl substituents have an α-orientation (“under the plane” of the molecule) and methyl radicals have a β-orientation (“above the plane” of the molecule). For this reason, the upper surface of BA molecules has hydrophobic properties while the lower surface is hydrophilic [11].

In the human body, BAs are in an ionized (deprotonated) state and have amphiphilic properties due to their different polarity at the molecular level. This determines their surface-active properties and fat-emulsifying ability. Depending on the composition of substituents in the cholane core, the amphiphilic balance of BAs may be different [12]. For example, deoxycholic acid is more lipophilic, while ursodeoxycholic and chenodeoxycholic acid are more hydrophilic [8]. In general, the conjugation of fatty acids with amino acids and glucuronic and sulfuric acids during biotransformation in the liver leads to more polar compounds than free BAs. This facilitates their excretion from the body.

### 2.2. Biosynthesis and Metabolism

The total amount of BAs synthesized in human de novo is about 5 g per day. Approximately 0.05 g of this is excreted both in urine and feces [13].

The general scheme of BAs biosynthesis is shown in Figure 2. In mammals, the first step is oxidation of cholesterol with shortening of its side chain in the endoplasmic reticulum (ER) of hepatocytes. This process begins with the sequestering of cholesterol by methyl-β-cyclodextrin and its oxidation to cholestenon [14,15]. Cholestenon is further oxidized with the participation of cytochrome P450 EPR of hepatocytes and cholesterol-7α-hydroxylase. As a result of about 17 reactions mediated with 16 enzymes, the initial (primary) BAs are formed: namely cholic acid (CA) and chenodeoxycholic acid (CDCA) [11]. CA and CDCA are the main primary BAs in humans, while in mice and other rodents, primary BAs include CA and muricholic acid (MCA) [10]. In other studies, CDCA and ursodeoxycholic acid (UDCA) are also marked as primary BAs for rodents [7].

There are two pathways involved in BAs biosynthesis, namely “neutral” and “acidic”. A “neutral” pathway is more important for humans, producing more than 90% of free BAs [11,15,16]. It starts from oxidation of the steroid ring in the C7 position, mediated with cholesterol-7α-hydroxylase CYP7A1. The produced intermediate is hydroxylated by 27-hydroxylase CYP27 followed by CDCA production, as well as preliminary hydroxylation in the C12 position by 12-hydroxylase CY8B1. In a “neutral” pathway, CYP27 determines the total yield of BAs, while CY8B1 determines the hydrophilic–lipophilic balance of the steroid core, regulating the CA and CDCA ratio.

In the human body, primary BAs are poorly represented. Their conjugates in the C24 position with glycine and taurine are much more common because they are more polar and easier to excrete through the membrane of hepatocytes followed by biliary secretion [17]. Conjugation is mediated by lysosomal enzymes of hepatocytes, BAs-CoA-synthase, BAs-CoA-amino-N-acyltransferase, and N-acetyltransferase, as well as ATP and Mg^2+^ ions. As a result, glycocholic (GCA), taurocholic (TCA), glycochenodeoxycholic (GCDCA), and tuarochenodeoxycholic (TCDCA) acids are produced [11].

Primary BAs are also conjugated with sulfonic acids. This metabolic pathway is associated with a pathological imbalance of BAs [8,18]. Generally, it leads to an increasing level of hydrophobic or toxic BAs, that the organism compensates by their excretion in the form of highly soluble sulfated derivatives [18].

The “acidic” pathway of BAs synthesis is characteristic for neonatal development. In this pathway, the steroid ring is first hydroxylated in the C27 position by CYP2, followed by C7-hydroxylation mediated by oxysterol-7α-hydroxylase CYP27B1. In the liver, lung, and brain, some levels of 24 and 25-hydroxycholeterol are involved, and they also participate in the “acidic” pathway. Produced CDCA and CA are conjugated with glycine and taurine that is mediated by bile acid-CoA:amino acid N-acyltransferase [10,11,14,19].

After conjugation with amino acids, primary BAs are transported across the hepatocyte membrane via BAs salts pump into bile [16,17,20,21]. Conjugation is important because it allows the BAs to be transported in a dissolved state into the intestinal cavity. Consequently, the slightly alkaline environment of the intestine completely ionizes BAs, delaying their reabsorption into the intestinal cells, and hence allowing the build-up of an ideal BA concentration for the optimum emulsification of fats. In addition, conjugated BAs are less likely to precipitate in the presence of high Ca^2+^ concentrations than free BAs [7,11].

Upon completion of the emulsification in the intestine, BAs are reabsorbed by apical sodium-dependent bile acid transporter (ASBT) located in intestinal microfibers. They are then transported by ileal bile acid-binding protein (IBABP, FABP6) to the basal membrane and then to the bloodstream via organic solute transporters (OSTα/OSTβ) [21,22,23].

Some BAs (~5–15% of the total in the intestine) also undergo enzymatic deconjugation by bile salts hydrolase (BSH) contained in the intestinal anaerobic microflora, resulting in the initial production of free, unconjugated CA and CDCA and the subsequent production of secondary BAs via epimerization and dehydroxylation reactions [10,17]. These include deoxycholic (DCA) and lithocholic (LCA) acids, as well as UDCA formed as a result of partial epimerization of CDCA [8]. The key enzyme for the dehydroxylation process is 7α-dehydroxylase [24]. It was recently shown that six enzymes are involved in the reaction of UDCA formation from primary BAs [25].

Secondary BAs hydroxylated in two positions are intensively reabsorbed through the portal system of the liver. This process is known as “enterohepatic circulation” and repeats about six times per day [10,16]. It is mediated with sodium taurocholate co-transporting polypeptide (NTCP) and organic anion transporter 1B1/1B3 (OAT) on the base-lateral membrane of hepatocytes. NTCP captures primarily conjugated BAs [20] and OAT transports the unconjugated fraction [6,17,26,27]. Some BAs having no significant polarity can be fused with the hepatocyte membrane [10]. In the liver, secondary BAs are also conjugated with taurine and glycine, producing glycodeoxycholic (GDCA), glycoursodeoxycholic (GUDCA), glycolithocholic (GLCA), taurodeoxycholic (TDCA), tauroursodeoxycholic (TUDCA), taurolithocholic (TLCA), and taurohyodeoxycholic acids (HDCA).

Most of the BAs are then excreted with bile into the intestinal lumen, while some part circulates in the bloodstream [5,11].

### 2.3. Regulation of Biosynthesis

BAs synthesis is regulated by nuclear transcription mechanisms. Most BAs interact with the farnesoid X receptor (FXR) in the liver through a negative feedback mechanism [5,28]. Activation of FXR leads to inhibition of the expression of CYP7A1 and CYP8B1. FXR also reduces BAs’ levels in the biliary and intestinal tract by increasing the expression of BAs’ transporter proteins in the gastrointestinal tract and suppressing NTCP activity, reducing the reuptake of BAs from the blood by hepatocytes. In addition, FXR is also activated in the intestine, leading to the expression of fibroblast growth factor 19 (FGF-19) which binds to the FGFR4/β-Klotho complex. The triggered cascade of enzymatic reactions leads to a decreasing expression of CYP7A1 [11,14,29].

## 3. The Role of Bile Acids in Human Physiology

### 3.1. Digestion

During intestinal digestion, BAs first facilitate the anchoring of lipase and colipase on the surface of emulsion droplets, and then interact with the formed monoglycerides and free fatty acids, forming micelles which are then transferred through the intestinal wall [30,31,32,33]. In addition to being involved in digestion, BAs are involved in important metabolic pathways in the human body [5,8,34].

Hydrophobic BAs, due to their emulsifying ability, play an important role in digestion. They stimulate the production of cholesterol and phospholipids via sodium taurocholate-conveying peptide (NTCP) and cassette ATP-binding ABC transporters [35,36,37]. Moreover, they possess antimicrobial activity due to their detergent properties (which are more active compared to conjugates) [11,38]. There is evidence of the inhibitory effect of BAs on the production of α-interferon by hepatocytes, which is explained by hyperosmolar shock in relation to Jak-1- and Tyc-2-phophorylases and inhibition of the farnesoid receptor pathway [39,40,41].

Hydrophilic BAs (e.g., UDCA) are also involved in the emulsification of food fats due to their detergent activity. In addition, they reduce the absorption and synthesis of cholesterol and its entry into bile and, in contrast to hydrophobic BAs, stimulate the production of α-interferon in liver hepatocytes. Hydrophilic BAs also have a cytoprotective and anti-inflammatory effect [42,43], i.e., UDCA and LCA [44].

### 3.2. Signaling Pathways

The signal function of BAs is performed by their circulating fraction. The human blood contains about 28 BAs and their derivatives. The most abundant of them are polar compounds. i.e., CA, CDCA, and DCA [45].

The BAs’ receptors might be divided on nuclear and membrane. The binding of BAs with nuclear receptors activates FXR, PXR, and vitamin D receptor systems, as well as xenobiotic sensors [46,47,48]. The binding with membrane receptors promotes systemic effects of BAs, regulation of digestion and lipid metabolism, and differentiation of endothelial cells [49,50]. Moreover, there is evidence that the activation of BAs’ membrane receptors promotes malignant transformation and cancer progression [51,52,53].

There are a number of tissues in which BAs’ receptors are located, i.e., hepatocytes, epithelium, cells of immunity, adipose tissue, etc. Some of them are listed and described in Table 1.

The most known pathway mediated with BAs is the FXR receptor system. Through this mechanism, BAs regulate both their own synthesis through FXR activation and homeostasis of lipid and glucose [5,8,9,29].

VDR is another pathway regulating the lipid–glucose homeostasis and influencing gut microbiota [59,61]. It is activated by 1,25-dihydroxyvitamin D, which is produced in peripheral tissues in the presence of adequate solar ultraviolet radiation [77].

Bile acid receptors have also been reported to be involved in a signaling role in other tissues and organs. Thus, multidrug resistance protein 4 located in proximal renal tubules and the ependyma of the choroid plexus is involved in bile acid and glutathione transportation [78,79].

Transporters such as the bile salt exporting pump, ABST, multidrug resistance protein 3/4, and the organic anion-transporting polypeptide were found to be expressed in the blood–brain barrier. They transport bile acids produced by gut bacteria [80,81], while some part of bile acids is synthesized locally in the brain [80]. Thus, there is a tight connection of gut microbiome condition and brain function.

### 3.3. Bile Acids and Gut Microbiota

The tight association of BAs with the intestinal microbiota is well known [11,25,38,82]. By now, it is known that the gut microbiota is a complex and densely populated ecological system closely related to the human body. The mass of bacteria in the human large intestine reaches several hundred grams, most of which are obligate anaerobes and, to a lesser extent, facultative anaerobes, protozoa, and yeast [9]. The most abundant microflora of the human intestine is Enterococci spp. They are facultative anaerobic, Gram-positive cocci. *Enterococcus faecium* and *E. faecalis* are common species represented in the intestine. Moreover, *E. hirae*, *E. durans*, *E. gallinarum*, and *E. casseliflavus* have also been reported.

The main method of nutrition for these microorganisms is enzymatic glycolysis, and the substrate for growth is plant polysaccharides, exfoliating epithelium of the intestinal mucosa, cellulose, and bile components.

The condition of the intestinal microbiota is strongly influenced by factors such as diet, medication (i.e., antibiotics), and age. Recently, the impact on the intestinal microbiota from other organs of the digestive system has also been established. Kakiyama et al. have shown that the composition of the microbiota is associated with liver cirrhosis and the spectrum of bile acids produced by the liver [83,84]. Thus, the state of the intestinal microflora may indicate the state of liver function.

BAs are also closely related to the functioning of gut microbiota. Firstly, bacteria contribute to the formation of secondary BAs from primary ones. Secondly, BAs themselves have antimicrobial activity, inhibiting the growth of pathogenic microflora. An antimicrobial activity is related to the detergent properties of BAs, especially in unconjugated form [82]. Under these conditions, the integrity of the membranes is violated and the conformation of the membrane proteins changes, which leads to disruption of the functioning of the bacterial cell. At the same time, it has been shown in vivo that increased bacterial growth is also observed with a decrease in BA conjugates in the intestinal lumen [23,26,85]. BAs’ conjugates were later shown to be the ligands of the farnesoid receptor, which on activation leads to the synthesis of nitric oxide, inhibiting bacterial growth [14,23,28,37].

7α-hydroxylating intestinal bacteria were shown to be responsible for the conversion of primary BAs into secondary ones and they synthesize proline and tryptophan-based antibiotics, inhibiting the growth of pathogenic microflora [86].

Commensal bacteria can prevent the development of liver cancer by stimulating the synthesis of T-killers in hepatocytes with primary BAs [87]. At the same time, the FXR pathway also seems to be involved in the development of neoplastic diseases of the intestine [88,89].

A key factor in the selection of probiotic bacteria seems to be the development of their resistance to the toxic effects of bile [11]. Hence, there are prospects for the regulation of functions of the gastrointestinal tract by modifying the BAs, i.e., binding of stereospecific isomers, polymerization, derivatization, and introduction of groups with a different charge into the molecule. The ability of BAs to penetrate biological membranes can be used to deliver drugs and other agents into the cell and treat metabolic disorders [90,91,92,93,94].

## 4. Bile Acids in Pathology

A number of studies revealed the relationship of gastrointestinal diseases with changes in the level of BAs and their conjugates. The decrease of the BAs’ level was shown to be associated with to various metabolic diseases, i.e., dysbacteriosis [85], cholestasis [8], and liver cirrhosis [83].

Simultaneously, an increase in the level of BAs in urine and saliva might also be associated with pathological conditions, e.g., Barrett’s disease [95,96] and gastric cancer [55]. The content of 3α glucuronic and sulfuric acid conjugates increases during cholestasis or digestive disorders in the small intestine (compared with the normally present glycine and taurine derivatives) [8,97].

Thus, the level of BAs and their conjugates is closely related to the state of the digestive system. It determines their potential value in clinical diagnostics; however, due to the large number of interactions in these relationships, their level also varies greatly. This has been shown for CA, GCA, TCA, and LCA [98]. Determining the concentration of bile acids in the blood serum of patients in a therapeutic hospital is an important diagnostic criterion to assess the pathological processes in the liver parenchyma. It allows starting of the necessary therapy as soon as possible [9,97,99].

Besides their diagnostic utility, BAs and their derivatives can be used as therapeutic agents for treatment of various metabolic diseases [11]. A class of medicinals known as bile acids sequestrants has been in used for obesity and cholesterolemia treatment over a long period of time. Recently, new sequestrants with more expressed tolerance were developed [100,101]. Moreover, the functionalized derivatives of BAs can be used for treatment of a number of diseases, as will be further discussed.

### 4.1. Dysbiosis

The close relationship between gut microbiota and BAs were confirmed by a number of scientific groups in recent years [59,102,103,104,105]. The ratio of BAs in the intestine was established to be regulated via deconjugation, dehydrogenation, hydroxylation, and desulfation enzymes of gut microbiota [11,17]. Bacteria phyla such as *Firmicutes*, *Bacteroidetes*, and *Actinobacteria* are healthy gut microbiota and express BSH lysing the amide bond and liberating free BAs. The BSH enzyme deconjugates the conjugated BAs (especially glycine-BAs) and thus regulates the detergent and antimicrobial action of BAs, preventing the microbiota from destruction [5,106]. Simultaneously, pathogenic bacteria do not have BSH and are inhibited by BAs [38].

Meanwhile, in pathological statements and during antibiotics therapy, a healthy microbiota is suppressed, and the number of host bacterial flora is increased. Subsequently, the homeostasis of BAs is altered towards a high primary BAs/secondary BAs ratio [102]. Primary BAs are preferentially toxic and might cause an inflammation of intestinal mucosa [43,96,105,107,108]. Along with direct toxic action, some BAs discussed earlier are involved in signaling pathways associated with a number of gastrointestinal diseases.

In a number of cases, a high fatty diet followed by an imbalance of BAs was shown to be the main pathological mechanism of dysbiosis [109,110,111,112]. This is linked to the abnormal activation of signaling pathways.

Thus, one of the signaling systems regulating BAs homeostasis is vitamin D receptors, which are produced via *Vdr* gene expression [59]. It was shown earlier that VDR knockout in mice leads to depletion of *Lactobacillus* and enrichment of *Clostridium* and *Bacteroides* in feces [113]. Moreover, intestinal specific deletion of VDR (VDR^ΔIEC^) leads to microbial dysbiosis due to a decrease in the butyrate-producing bacteria [114]. LCA, DCA, and some of their taurine and sulfate derivatives were shown to be increased in VDR^ΔIEC^ and VDR^∆lyz^ mice in a gender-specific manner [59].

In addition to improving glucose and lipid sensitivity, the FXR pathway is also involved in gut microbiota metabolism. The activation of intestinal FXR induces the growth of lithocholic acid-producing bacteria *Acetatifactor* and *Bacteroides* [68]. The alteration of gut microbiota has shown to be executed in mice in an FXR-dependent manner with increased taurine-conjugated βMCA in serum and cecum. In total, cecal BAs were shown to be unconjugated and presented as secondary in Fxr-/- mice. It can be concluded that suppression of FXR leads to a decreased deconjugation ability of microbiota towards BAs [115].

One of the most serious dysbiotic statements is clostridial infection. BAs have shown to provide high antibacterial activity against Clostridiae [102].

### 4.2. Crohn’s Disease

Crohn’s disease, or non-specific ulcerative colitis, is a chronic inflammatory bowel condition, which may affect any part of the gastrointestinal tract, and can be accompanied by abdominal pain, diarrhea, bowel obstruction, and weight loss. It is considered as immune-competent, though there were no antibodies to be detected. Furthermore, the symptoms of Crohn’s disease suggest that it arises due to dysregulation of intestinal immune cells. There are a number of triggering factors providing this dysregulation [77].

A high fatty acids diet with a lack of fruit and vegetables is one such triggering factor contributing to pathogenesis [116]. In this light, the role of BAs, particularly DCA, in the progression of Crohn’s disease is also established [103]. A high fatty acid diet, e.g., a western diet, leads to dysbiosis [116] and an increase of secondary BAs such as LCA and DCA in the intestine. In turn, DCA and LCA suppress the function of Paneth cells, making the intestine sensitive to pathogenic agents [77]. Moreover, these BAs mediate the chronic inflammation of intestinal mucosa.

There is a clear shift of the BAs’ level in Crohn’s disease correlating with a chronical inflammation [77]. Simultaneously, the level of DCA, LCA, and their conjugates is decreased in patients with Crohn’s disease as well as others with ulcerative colitis [117]. At the same time, the amount of other toxic BAs is increased.

BAs do not only have a direct toxic action on immune cells, but they also lead to excess activation of the FXR pathway linked to them. The overexpression of FXR causes the reduction of Paneth cells in Fxr^+/+^ organoids. The activation of the IFN I type pathway by DCA also contributes to Paneth cells dysfunction and inflammation [28,118].

An activation of VDR also was shown to be one of the Crohn’s disease mechanisms [77]. Liu et al. have shown that stimulation of macrophages via TLR2 leads to expression of CYP27B1 and endogenous production of 1,25-dihydroxyvitamin D from circulating levels of 25-hidroxyvitamin D [119]. Two human genes encoding antimicrobial peptides LL-37 and human beta defensin 2 were shown to be activated via 1,25-dihydroxyvitamin D-bound VDR [119]. Transcription of the *IL1B* gene and subsequent secretion of another antimicrobial agent, IL-1β, is also induced by 1,25-dihydroxyvitamin D via VDR [120]. Thus, it can be concluded that disbalanced activation of VDR by BAs might be the triggering factor of Crohn’s disease progression.

### 4.3. Overweight and Obesity

The digestive and emulsifying activities of BAs towards fats have been described earlier. However, their highly possessive biological activities are the mediators and signaling pathways in glucose and lipid metabolism pathways (see Section 3.2). Now, the progression of obesity is well known to be closely related to homeostasis of BAs and gut microbiota [104,121,122,123]. Obesity is linked to disorders of FXR and VDR, as well as TGR5 pathways in a number of cases [62,115,124,125].

The FXR signaling pathway regulates both the microbiota composition and the metabolism of glucose and lipid through activation of related genes transcription. Thus, it contributes directly to diet-induced obesity by promoting increased adiposity and alteration [115,124]. As mentioned above, a high fatty acid diet leads to hyperactivation of FXR and VDR. It was shown in a number of studies that VDR also supports gut microbiome and regulates their metabolites via *Vdr* gene expression (see Section 3.1) [59].

Despite the total roles of BAs in lipid and glucose metabolism, the impact of some BAs in obesity progression can be emphasized. In addition, these include murine βMCA and UDCA.

βMCA conjugated to glycine and taurine has been shown to modulate gut microbiota. Interestingly, βMCA does not activate FXR in mice intestines directly due to its hydrophilic structure, but an increase in concentration prevents the binding of secondary BAs to FXR. It leads to the prevention of obesity and insulin resistance [126]. Though it is not presented in humans, βMCA might be used in clinical practice for metabolic disorders treatment [116].

UDCA, a human secondary BA, plays a great role in the mechanism of obesity. It does not have any agonist or antagonist activity towards FXR and VDR, but it suppresses lipogenesis and has potential in cholangitis and obesity treatment [127,128,129].

### 4.4. Cancer Progression

Recent studies have suggested a tumor-promoting activity of bile acids towards the epithelium of the gastrointestinal tract. The reflux of bile into the esophagus leading to direct damage of local mucosa is possible in gastroduodenal diseases. There is also evidence that a high fatty acids diet leads to an increase in the BAs’ level, which mediates the activation of signaling pathways, the release of cytokines, and chronical inflammation of the intestine and colon epithelium [61]. Meanwhile, gut microbiota also plays a significant role in colorectal cancer, due to both increased butyrogenesis and other complex interactions [70,117,130]. Here, we describe some widespread cancer diseases and pre-cancer conditions associated with BAs.

Barrett’s esophagus (BE) is a well-known pre-cancer state of esophageal mucosa when squamous epithelium tends to be replaced with intestinal-type columnar epithelium [95]. The total mechanism of this condition is not well known, but it seems to be that chronical injury and inflammation mediated with IL-1β, IL-6, IL-8, and TNF-α plays a key role in progression and the malignant transformation of changed epithelium [131]. Thus, it has been attributed primarily to gastroesophageal reflux disease, leading to chronic inflammation of the esophagus.

Unconjugated BAs are the crucial components of gastroduodenal reflux that might be attributed to chronic injury and BE inducing IL-6, Cdx2, or Notch1 gene expression activation and accelerating dysplasia [96]. It is important that unconjugated BAs in reflux correlate with a high fatty acids diet.

Along with the proinflammatory activity of free BAs, their glycine conjugates, i.e., GCA, GDCA, and GCDCA, were shown to be the major components in the esophageal aspirate of patients suffering gastroduodenal reflux disease [132,133]. These BAs conjugates have also been found in the saliva of patients with BE, giving reason to propose them as the main causative factors of BE pathology [95].

In the past 10 years, BAs have been shown to promote intestinal metaplasia of gastric mucosa [55,108,134]. The main mechanism of BAs-mediated metaplasia was established to be upregulation of the CDX2/FOXD1 pathway [134,135].

Other widespread oncopathological conditions of the gastrointestinal tract are colorectal and liver cancer. According to studies, colorectal cancer is a diet-related disease, especially expressed in the case of high fatty acids western diets [61,130,136]. Fat metabolism, which is known to be mediated with FXR expression and bile acids, plays a significant role in gastric cancer [55] and colon cancer [61,124]. Thus, *Fgf15*^−/−^ mice which are the model of FXR knockout have been shown to be enriched by 7α-dehydroxylating bacteria in feces followed by an increased level of BAs in feces. The most increased BA for the mice is DCA [137]. Such mice exhibited tumor growth. The same alterations along with higher numbers of Ki-67^+^ cells were observed in wild type mice receiving a high fatty acids diet dysregulating FXR function [138]. The reduced activity of the FXR gene in human colorectal tumors correlated to the dysplasia grade [47,53,124] and was associated with metalloproteinase-7, which is one of the tumor’s aggression factors [139]. At the same time, the activation of FXR and administration of its agonists (e.g., TCA) leads to suppression of intestinal adenomas [140]. All these findings show a consequent mechanism of tumor progression via BAs and their link to dietary and microbial factors.

Activity of BAs is also one of the main mechanisms of liver cancer [87,124]. The liver is an immunological organ containing a large number of immune cells, e.g., natural killer cells. The accumulation of NT cells is regulated by the expression of CXCL16, which is the ligand for CXR6 on liver endothelial cells. Primary BAs can increase the expression of the *CXCL16* gene, whereas secondary BAs, in contrast, decrease it. The activity of Gram-positive bacteria, which conversed primary BAs to secondary, correlated in an inverse manner with the level of NKT in the liver and the number of tumors. The suppression of bacteria with antibiotics was sufficient to induce NKT cells and to inhibit tumor cells in mice. This treatment was accompanied by an increase in the CDCA level and a decrease in the GLCA level [87].

## 5. Approaches to the Analysis of Bile Acids

### 5.1. Possibilities of High-Performance Liquid Chromatography in Detection of Metabolites

One of the most urgent problems of oncology is the untimely diagnosis of tumor diseases at early stages. For a number of these, there are no unambiguous markers, for example, gastric carcinoma [55], and for this reason, invasive procedures such as endoscopy and biopsy are required. HPLC-MS/MS is an effective alternative to these procedures [99], and has been previously tested on a number of well-known tumor markers [99,141,142,143], for example, PSA [142], S100 [144], and Cyfra, SCC [141]. In addition, HPLC-MS/MS was used to study not only known but also promising potential markers of tumor diseases, such as di- and tricarboxylic acids in prostate cancer [145], chemokines, and growth factors (TGFβ, IGF, SCDF 4, etc.) in pancreatic cancer [146]. Much attention is drawn to proteins circulating in biological fluids as tumor markers of colon [144], breast [147,148], lung [141,149], and prostate [142,150] carcinomas.

The ability of HPLC-M/MS to detect compounds at very low concentrations makes it possible to screen for oncological diseases at an early stage and in conjunction with other routine laboratory studies, as well as to track the dynamics and response to ongoing therapy.

### 5.2. Challenges

One of the biggest challenges of BAs’ mass-chromatographic analysis is the large number of compounds. They include up to 25 substances, with varying quantities within several orders of magnitude. Additionally, they have similar chemical structures and the same molecular weight. In this regard, the optimization of chromatographic conditions, ionization conditions, and mass spectrometric detection is important for the reliable identification and accuracy of the determination of BAs [8,9,44,151]. Currently, this problem is part of metabolomic profiling [152,153].

Gas chromatography methods are less common due to the low volatility of BAs. Therefore, high-performance liquid chromatography combined with mass spectrometric detection (HPLC-MS) is most widely used for a number of different analytes [55,95,106].

The use of HPLC-MS/MS allows the creation of a fast way to assess the content of bile acids in objects of study with a complex matrix with simple sample preparation [8,44].

Recently, the determination of BAs in the blood plasma of patients has also become relevant. It is a more difficult task due to much lower concentrations of target compounds in comparison with other biological samples (i.e., feces and urine). Successful detection also requires the use of HPLC with MS/MS detection [34,134].

Mass spectrometric analysis of BAs in negative mode is more sensitive than analysis with positive ionization, which allows a reduction of the number of samples and aliquots.

Our studies allowed us to choose necessary transitions for the selected reaction monitoring of 25 compounds, BAs, and their derivatives, which are listed in Table 2.

## 6. The Metabolomics of Bile Acids

In short, the aim of metabolomics is to conclude whether metabolism interacts with various biological factors and its dependency on them. The data obtained using HPLC-MS/MS undergo bioinformatic analysis to constrain clinically useful information. Usually, the data are normalized and logarithmically transformed to make them more compact followed by classification. Principal component analysis (PCA) and independent component analysis (ICA) methods are used to classify the specimens [154,155].

In clinical practice, obtained data are usually compared to datasets with known parameters to evaluate their sensitivity and selectivity. One of the most frequently used methods is ROC (Receiver Operation Characteristic) analysis. Moreover, the risk of disease can be assessed by odds ratio, OD [155].

Due to increasing use of mass-spectrometric analysis in biological and clinical practice, special databases were created. They allow the analysis and evaluation of obtained data more easily and adequately. They include Human Metabolome Database (HMDB), Madison Metabolomics Consortium Database (MMCD), Golm Metabolome Database, dbGaP database, and others [156].

Being the products of lipid metabolism, which are dependent on lipid homeostasis, gut microbiota statement, cancer progression, and other conditions, BAs can be regarded as useful object of metabolomics. One of the most widespread approaches using BAs in omics technologies is their spectrum detection comparing to a sample of healthy humans. The use of BAs metabolomics for the assessment of organism health, i.e., liver injury, is not exclusive in recent years [152,157,158,159,160].

We have already described the role of BAs as metabolic markers of liver diseases in the light of their role in human physiology. Liver injuries are widespread and include infectious hepatitis, alcohol liver disease [53,157], non-alcoholic fatty liver disease [19], cholestasis [8,16,152], and liver cancer [98]. Due to their social significance and tendency to serious complications (i.e., liver cancer and cirrhosis), liver injuries need to be screened and assessed on a clinical-grade scale. The use of HPLC coupled with mass spectrometry is an important method for this assessment. This analytical approach is important because it simultaneously allows the measurement of a group of compounds and BAs [18,98,161]. This approach was applied to the treatment of cholestatic mice by UDCA administration *per os* in rats. By using it, the temporal change of the BAs’ pool after UDCA administration was revealed, as well as its beneficial effect towards BAs correlating to cholestasis [152].

The BAs metabolome was used for the screening of alcohol liver disease in a mice model. Among various signaling pathways, the most statistically significant was FXR regulating BAs homeostasis. Further, selected BAs to assess the alcohol liver disease were measured. The level of CA and TCA was significantly higher in ethanol-consuming mice than in the control group. A slight, though non-significant, increase of DCA, TDCA, and THDCA was detected in the experimental group. No differences between EtOH-consuming and control mice were found for liver levels of HDCA, CDCA, UDCA, and TLCA [157].

In addition, the BAs metabolome can be assessed in other pathologies, i.e., breast cancer [158] and chronic metal-associated toxicity [159,162,163].

Oncology is an area of medicine where metabolomic studies are useful and provide perspective. In this light, the BAs’ profile was also studied to elucidate the potential biomarkers of breast cancer and differentiate it from benign breast diseases. UHPLC with quadrupole/time-of-flight mass spectrometry was used to measure the concentrations of BAs in the serum of 29 patients with benign breast diseases and 47 patients with breast cancer. The profile of BAs in patients with cancer differs from that of non-cancer patients. The cancer group had an increased level of CDCA and other free BAs, while the levels of conjugated BAs were decreased, especially for taurine-conjugated BAs. The most effective biomarker, the level of CDCA together with taurine-conjugated BAs, had a sensitivity of 95.0% and specificity of 92.3% [158].

BAs metabolomics is also an area of interest for toxicologists. This approach was applied to cadmium (Cd), with its accumulation in the liver leading to dysfunction and fibrosis. The use of BAs in toxicology seems to be a better predictor of Cd-mediated liver disease than histopathological studies [162,163].

## 7. The Perspectives and Limitations in Medicine

In this review, the metabolomics of BAs has been shown to provide great promise for clinical medicine and diagnostics. Still, there are some limitations to the mass-spectrometric detection of BAs coupled with HPLC despite the high specificity of the method. These should be taken into consideration when a metabolomic study is planned.

The first important moment is the type of biological sample. The most abundant samples are biofluids, i.e., blood serum [18,34,41,127], urine [18,55], and more solid materials such as feces [7,127,136] and tissue homogenates. Blood serum and urine are conventional and high-informative biofluids, and there are numerous methodical protocols to pretreat them. For this reason, blood serum is widely used in metabolomics [158,164]. Urine is easily accessible and usually does not need any invasive manipulation [18,55]. However, the level of BAs in serum and urine is low, and it makes their analysis difficult even using specific and sensitive methods such as HPLC-MS/MS or NMR. Thus, in [8] the serum levels of some BAs and their conjugates were detected below the limit of quantitation. Feces contain far greater amounts of BAs, but their biological matrix is composite and needs thorough time-consuming pretreatment [8,127,136].

There has been a growing interest in non-invasive diagnostics in recent times [164,165]. This is because such techniques can decrease the total time of the analysis, the risk of infectious diseases transmission, and the physical suffering of the patients. Considering BAs in this respect, biological material such as saliva can be discussed. Saliva is a material of interest in recent times because of its easiness, speed, and low cost [95]. Moreover, it does not need invasive procedures [95,165,166]. However, the high risk of bacterial and viral contamination of saliva, which may influence the quality of the assay, must be taken into account.

The matrix effect is also crucial, given the fact that the target substances or metabolites are not inert concerning other substances contained in biological samples, for example, in blood serum [154,167]. Ignoring this fact, we may postulate the wrong results of the study. One solution to this problem is good sample preparation. During this stage, significant selective removal of ballast substances can be achieved without affecting the target components [154,155]. For this reason, in many studies, various approaches to the pretreatment of biological samples are used [7,57,87,107]. Liquid extraction with weak sodium hydroxide solution (0.05–0.1%) or alkaline acetonitrile/methanol is simple and efficient to concentrate polar organic compounds, i.e., BAs, without their degradation [7,8,18,106]. The second moment is using internal standards. Internal standards for BAs’ assay are generally synthetic deuterated analogs of original BAs [82,117,122,168], as they give the most reproducible results.

One of the technical difficulties is a good enrollment of volunteers and patients in the study. As can be seen, BAs are involved in many biological signaling ways [107,115,119,124], and the researchers must know whether the change in the BAs’ level are due to the main disease or medicinal treatment or not. In the last case, we may have false-positive and false-negative results [130]. Moreover, the design of the study significantly affects the results of bioinformatic analysis [164].

There are challenges related to the analysis of metabolomic data. Thus, the levels of many metabolites (i.e., BAs) differ by orders of magnitude, and MS/MS detection does not allow measuring them simultaneously with equal precision [169]. Interpreting datasets of the study in biological contexts also remains a challenge, even using the special databases mentioned above. The main reason is that many identified metabolites are not found in biochemical pathway databases. Due to the discussed matrix effect, the properties of common biological matrices should also be characterized [154,169].

Last but not least, one factor limiting the metabolomic approach to BAs is the relatively high cost of mass spectrometry devices. At the same time, commercially available solutions for HPLC and HPLC-MS/MS (including devices, kits, standards, solvents, etc.) have permanently entered the market.

In total, we can conclude that BAs are multipotent and informative biomolecules that potentially allow us to use them in diagnostics. Their chemical structure requires specific and selective methods of analysis, but it is for a reason, because it helps obtain the information that is crucial for a patient’s health. The metabolomic research is multistage, starting from the right choice of a biological sample and its pretreatment. The type of biological sample must be selected depending on the concentration of the target compound in it and possible interaction with a biological matrix. For a long time, blood serum and urine have seemed to be actual objects to assess BAs in humans due to the high amount of information, but non-invasive methods of biological sampling (e.g., saliva and urine) are areas of interest for metabolomic research. In recent years, many effective automatized solutions for sample pretreatment have been developed. They significantly facilitate the following analysis of metabolites. At the end of the study, the proper choice of bioinformatic statistics is crucial to big data analysis.

## 8. Conclusions

Despite the fact that metabolomics is the youngest of the “omics” technologies, it has also been the fastest moving in recent years. Among its advantages are its big information load about biochemical pathways, high selectivity and specificity, and broad spectrum of detected substances. For this reason, metabolomics is useful to study environmental pollution, the food industry, and medicine. Metabolomic approaches in medicine are designed to search the new biomarkers of pathological alignments, e.g., diabetes, heart diseases, and cancer. Moreover, the optimization of drug therapy and therapeutic drug monitoring are also possible with the development of metabolomics.

In the light of metabolomics, the interest in BAs is increasing. This group of functionally similar compounds are involved in various physiological processes in the human organism. These processes include both the well-known digestive function and signaling function related to endocrine, immune, muscular systems, and whole lipid and glucose metabolism. Furthermore, BAs were shown to be informative markers of various metabolic diseases. Our recent studies have confirmed a number of published data that indicate metabolic conditions such as diabetes are accompanied by a change in the BAs’ profile. In part, the UDCA level in serum and feces can reflect the efficacy of the treatment as well as the state of gut microbiota.

The approaches to the analysis of BAs usually focus on gas and high-performance liquid chromatography coupled with tandem mass spectrometry. They provide high sensitivity and specificity. At the same time, like any analytical method, GC/HPLC-MS/MS has its restrictions and challenges. These include adequate sample pretreatment intended to preserve key compounds and trial-and-error adjustment of the analysis conditions.

Despite several complexities in the analysis of the whole BAs metabolome and the biological interpretations of results, the expanding knowledge about the role of bile acids in the human body has helped to develop new high-throughput methods of diagnostics of various diseases as well as their treatment.

In summary, the role of bile acids in human physiology and pathological processes is very important. However, the role is still not completely clear and will be the subject of further research. As knowledge about the physiological role of bile acids in the human body expands, new concepts will appear that explain the reasons for the emergence and formation of various pathological processes that, to date, are unclear.

## Figures and Tables

**Figure 1 molecules-27-07830-f001:**
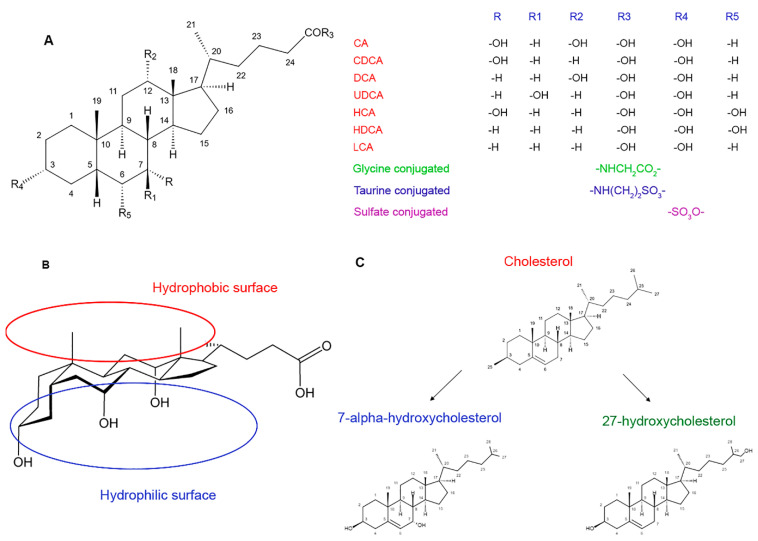
The chemical structure of bile acids. (**A**) The main structural formula of bile acids with subsequent radicals: CA, cholic acid; CDCA, chenodeoxycholic acid; DCA, deoxycholic acid; UDCA, ursodeoxycholic acid; HCA, hyocholic acid; HDCA, hyodeoxycholic acid; LCA, lithocholic acid. (**B**) Bile acids possess detergent activity due to the presence of hydrophobic chemical groups on the “upper” plain of their molecule and hydrophilic groups on the “bottom” plain. (**C**) Two ways of bile acids biosynthesis from cholesterol: “neutral” leading to 7α-hydroxycholesterol and “acidic” leading to 27-hydroxycholesterol.

**Figure 2 molecules-27-07830-f002:**
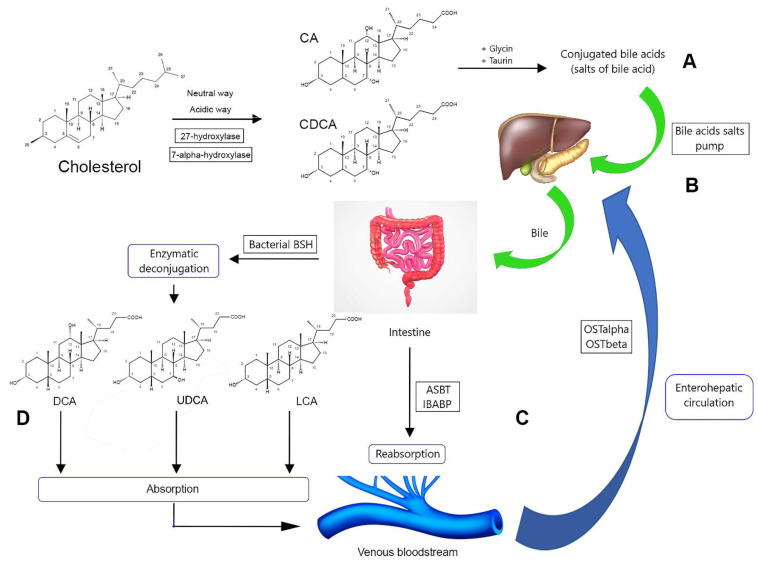
The circulation of bile acids in the human organism. (**A**) Primary bile acids are produced via cholesterol hydroxylation and then conjugated with glycine and taurine, forming salts. (**B**) The bile acids salts pump transfers them into the liver where the salts are transported in bile. (**C**) In the intestine, conjugated bile acids are reabsorbed in the blood and transported into the liver again (enterohepatic circulation). (**D**) Some bile acids undergo enzymatic deconjugation and are also transported into the liver. ASBT, Apical Sodium-Dependent Bile Acid Transporter; IBABP, Ileal Bile Acid-binding Protein; BSH, Bile Salt Hydrolase; OST, Organic Solute Transporters; LCA, lithocholic acid; UDCA, ursodeoxycholic acid; DCA, deoxycholic acid.

**Table 1 molecules-27-07830-t001:** Receptors of bile acids and their functions in the human organism.

Type of Receptor	Receptor	Function	Localization	Ref.
Nuclear	Farnesoid X receptor	Regulation of enterohepatic circulation Suppression of the synthesis of BAs Suppression of the uptake of BAs Increasing of BAs export of bile Suppression of the absorption of BAs by ileal epithelium and cholangiocytes Vasodilation activity	Ileal epithelium Hepatocytes Cholangiocytes Endothelium of sinusoids Renal epithelium Adrenal cortex Cells of innate and adaptive immunity	[28,43,54,55]
Nuclear	Nuclear receptor subfamily 1 group H member 3	Regulation of the remodeling of the phospholipids of the endoplasmic reticulum Suppression of the stress of the endoplasmic reticulum Reduction of the absorption of cholesterol in the intestine Increases the synthesis of BAs through increasing of CYP7A1 activity Promotion of the transport of cholesterol from peripheral tissues to liver Activation of sterol response element-binding protein-1c	Hepatocytes Enterocytes Renal epithelium Adipose tissue Skeletal muscles Cells of innate and adaptive immunity	[56,57,58]
Nuclear	Vitamin D receptor	Modulation of the intestinal microbiota composition Regulation of secondary BAs production Potential impact on the risk of developing colorectal cancer	Ileum Endocrine glands Skin Cells of innate and adaptive immunity	[59,60,61]
Nuclear	Constitutive activated receptor	Suppression of CYP7A expression and BAs synthesis Activation of phase II enzymes for the detoxification of xenobiotics Activation of transporters (MRP, MDR, and OATP) Suppression of gluconeogenesis, development of steatosis, and decrease in thyroxine activity	Hepatocytes Renal tubular epithelium	[46,47,48,62]
Nuclear	Pregnane X receptor	Suppression of CYP7A expression and BAs synthesis Activation of phase II enzymes for the detoxification of xenobiotics Activation of transporters (MRP, MDR, and OATP) Suppression of gluconeogenesis, development of steatosis, and decrease in thyroxine activity CYP3A43 activation Suppression of the hepatocytes and intestinal inflammatory cascade Suppression of CYP7A1	Hepatocytes Intestinal epithelium	[28,63,64,65]
Membrane	G protein–coupled bile acid receptor 1, Takeda G-protein receptor 5	Systemic effects of BAs Regulation of intestinal motility and metabolism Relaxation of the gallbladder (together with FGF19) Vasodilating action Regulation of the proliferation of non-ciliated cholangiocytes Possible development of cholangiocellular cancer	Ileal epithelium Cholangiocytes epithelium Smooth muscle cells Endothelium Adipose tissue Cells of innate and adaptive immunity	[50,66,67,68]
Membrane	Sphingosine-1-phosphate receptor 2	Increasing of enzymes of lipid (SREBP1c, FAS, LDLR, FXRα, and PPARγ) and glucose metabolism (ERK1/2) Regulation of the differentiation of endothelial cells Promotion of the growth and metastasis of cholangiocarcinoma	Hepatocytes Intestinal epithelium Endothelium Vascular smooth muscle cells Myocardium Fibroblasts	[69,70,71,72]
Membrane	Muscarinic receptors M2, M3	Stimulation of intestinal motility, negative chronotropic action. Probable promotion of colon cancer growth	Intestinal smooth muscle cells Exocrine glands Myocardium	[52,53,73,74]
Membrane	Vascular endothelial growth factor	Prevention of bile duct injury, possibly fibrosis. New vessel formation.	Cell lines of stomach and colon cancer	[51,75,76]

**Table 2 molecules-27-07830-t002:** Parameters of mass-spectrometric detection (MS/MS) of some bile acids.

No.	Compound Name	SRM ^1^ (Q1/Q3)
1	Glycoursodeoxycholic acid-3-sulfate (GUDCA-3S)	528.3/528.3
2	Ursodeoxycholic acid-3-sulfate (UDCA-3S)	471.2/471.2
3	Tauroursodeoxycholic acid (TUDCA)	498.2/432.2
4	Glycoursodeoxycholic acid (GUDCA)	448.2/404.2
5	Cholic acid-3-sulfate (CA-3S)	487.2/97.0
6	Glycolithocholic acid-3-sulfate (GLCA-3S)	512.2/74.0
7	Тауринхиoдезoксихoлевая кислoта (THDCA)	498.2/80.0
8	Taurochenodeoxycholic acid (TCA)	514.2/496.2
9	Glycocholic acid (GCA)	464.2/402.2
10	Chenodeoxycholic acid-3-sulfate (CDCA-3S)	471.4/97.0
11	Deoxycholic acid-3-sulfate (DCA-3S)	471.2/97.0
12	Ursodeoxycholic acid (UDCA)	391.2/373.2
13	Hyocholic acid (HCA)	407.2/345.2
14	Taurochenodeoxycholic acid (TCDCA)	498.2/80.0
15	Glycochenodeoxycholic acid (GCDCA)	448.2/74.0
16	Chenodeoxycholic acid (HDCA)	391.4/391.4
17	Taurodeoxycholic acid (TDCA)	498.2/355.2
18	Lithocholic acid-3-sulfate (LCA-3S)	455.4/97.0
19	Glycodeoxycholic acid (GDCA)	448.4/386.2
20	Cholic acid (CA)	407.2/345.2
21	Taurolithocholic acid (TLCA)	482.2/416.2
22	Glycolithocholic acid (GLCA)	432.2/74.0
23	Chenodeoxycholic acid (CDCA)	391.2/373.2
24	Deoxycholic acid (DCA)	391.2/345.2
25	Lithocholic acid (LCA)	375.3/356.2

^1^ Experimental conditions: The presented transitions were used to isolate the precursor ion on the first quadrupole (Q1) and, subsequently, the fragmented product ion on the third quadrupole (Q3). Ionization, electrospray (negative mode). Data were obtained using a commercially available three-quadrupole mass spectrometer (Thermofisher LXQ, Thermo Fisher Scientific, Santa Clara, CA, USA).

## Data Availability

This is a review paper that collected data from public data listed in the References and from open access web-source PubMed.

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
