# Peer review of "Bile Acids: Physiological Activity and Perspectives of Using in Clinical and Laboratory Diagnostics"

_molecules, 2022, doi:10.3390/molecules27227830_

Round 1
Reviewer 1 Report
This manuscript should be published after addressing the following concers.
1. Label the carbon numbers for BAs scaffolds. Unify the structural format in Chemdraw.
2. Please doublecheck the English in the manuscript, esp for grammar errors. L561-L564, etc.
3. The subtitle in 6 can mislead the readers and please revise properly.
4. Please make an annotation for SRM in Table 2.
Author Response
We sincerely thank for your positive response, detailed analysis and useful comments of our manuscript. We have considered all raised questions and hope, that it’ll help readers in better understanding of our article.

Reviewer 2 Report
The manuscript, basically, is well written and organised, but major amendments are needed before acceptance. The comments are listed as below:
1. Abstract
Line 5, “the of bile acids may support the lipid…” There is/are missing word(s) between ‘the’ and ‘of’. Please clarify it.
2. Introduction
It would be better for the authors to add paragraph(s) or a few sentences to briefly describe the structure and content (e.g. how many section(s), what is the content of each section) of the manuscript to let the readers familiar with what they are going to read.
3. Content of the manuscript
Before conclusions (Section 7), please add two more sections. One of the sections is about “Recommendations” which should include the comments of the authors about bile acids, its future research direction and potential development and application. Another section is about “Limitations” which should involve the limitations of this study.
Author Response
Thank you very much for the comments on our manuscript. We have carefully considered all your suggestions and in response have made some changes in the manuscript. These changes are outlined in attach file.

Round 2
Reviewer 2 Report
The revised manuscript is acceptable for publication.